# The Role of MicroRNA in the Regulation of Tumor Epithelial–Mesenchymal Transition

**DOI:** 10.3390/cells11131981

**Published:** 2022-06-21

**Authors:** Jing Feng, Shaofan Hu, Keli Liu, Guiyin Sun, Yiguo Zhang

**Affiliations:** 1The Laboratory of Cell Biochemistry and Topogenetic Regulation, College of Bioengineering, Faculty of Medical Sciences, Chongqing University, No. 174 Shazheng Street, Shapingba District, Chongqing 400044, China; 202019021043@cqu.edu.cn (J.F.); 20171901005@cqu.edu.cn (S.H.); 201919021016@cqu.edu.cn (K.L.); 2Chongqing University Jiangjin Hospital, School of Medicine, Chongqing University, No. 725 Jiangzhou Avenue, Dingshan Street, Jiangjin District, Chongqing 402260, China

**Keywords:** miRNA, EMT, Snail1/2, Twist, ZEB1/2

## Abstract

Consistently, the high metastasis of cancer cells is the bottleneck in the process of tumor treatment. In this process of metastasis, a pivotal role is executed by epithelial–mesenchymal transition (EMT). The epithelial-to-mesenchymal transformation was first proposed to occur during embryonic development. Later, its important role in explaining embryonic developmental processes was widely reported. Recently, EMT and its intermediate state were also identified as crucial drivers in tumor progression with the gradual deepening of research. To gain insights into the potential mechanism, increasing attention has been focused on the EMT-related transcription factors. Correspondingly, miRNAs target transcription factors to control the EMT process of tumor cells in different types of cancers, while there are still many exciting and challenging questions about the phenomenon of microRNA regulation of cancer EMT. We describe the relevant mechanisms of miRNAs regulating EMT, and trace the regulatory roles and functions of major EMT-related transcription factors, including Snail, Twist, zinc finger E-box-binding homeobox (ZEB), and other families. In addition, on the basis of the complex regulatory network, we hope that the exploration of the regulatory relationship of non-transcription factors will provide a better understanding of EMT and cancer metastasis. The identification of the mechanism leading to the activation of EMT programs during diverse disease processes also provides a new protocol for the plasticity of distinct cellular phenotypes and possible therapeutic interventions. Here, we summarize the recent progress in this direction, with a promising path for further insight into this fast-moving field.

## 1. Introduction

MicroRNAs (miRNAs), as an important group of noncoding RNAs, are 19 to 24 nucleotide long RNAs, arising from the selective processing of the stem-loop region of longer RNA transcripts. Clearly, miRNAs perform important biological functions in posttranscriptional regulation by acting on their cognate target mRNAs encoding proteins [1]. Further evidence suggests that miRNAs also exert another regulatory role for single-stranded RNA molecules by targeting 2–8 nucleotides in its seed sequence to the 3′-UTR specific action site of mRNA [2,3]. In recent years, it have reported to play a key role in tumor metastasis, becoming an increasingly important indicator of clinical prognosis and tumor markers [4]. Compared with normal tissues, an anomalistic expression of miRNAs in tumor cells has been proven in multiple aspects to promote the emergence, development, migration, and invasion of tumor cells [5,6,7,8]. Notably, such miRNA regulation has been widely integrated with gene regulatory networks, making it easier for evolution to generate complex, highly connected regulatory networks that respond forcefully to environmental, genetic, or random contingencies. Consequently, abnormal expression of miRNA is always accompanied by the development of cell transformation toward carcinogenesis, including a high metastatic and invasive ability [9].

It has widely been accepted that the highly invasive and metastatic ability of tumor cells is one of the vital problems in cancer treatment. EMT has also been increasingly described in tumor metastasis and invasion, especially EMT-related transcription factors (EMT-TFs) that regulate cancer development and progression, including the conventional Snail1, Slug, ZEB1/2, Twist, and other unconventional ones, such as Wnt family member 1 (WNT1)/β-catenin, transcription factor pairing-related homeobox 1 (Prrx1), MYC proto-oncogene (c-Myc), and thyroid transcription factor 1 (TTF1) [10,11,12,13,14]. On the one hand, multiple signaling pathways, such as TGF-β, MAPK, PI3K/Akt, PTEN, and Wnt, are activated in order to enable activation of the EMT program, which controls the expression of those downstream genes to acquire tumor metastasis, but this may also achieve inhibition depending on those aforementioned transcription factors in different biological contexts [15]. On the other hand, the regulatory modes by nontargeted transcription factors have been gradually reported, including EMT marker proteins E-cadherin, N-cadherin, and vimentin, as well as other non-marker proteins [5,16,17]. Overall, different miRNA families target discrete transcription factors due to their distinct conserved seed sequences, and thus play a biological role, acting as tumor-suppressing or tumor-promoting factors, in various types of cancers [18].

## 2. Generation and Mechanisms of Canonical miRNAs 

As a portion of small regulatory RNAs, miRNAs do not encode proteins but play a vital role in the posttranscriptional regulation, particularly, of eukaryotes by targeting the 3′-UTR of target gene transcripts to reduce the expression of cognate proteins and regulate cell biological processes. Each miRNA combines with relevant Argonaute proteins to form an RNA–protein complex, which recognizes multiple target mRNAs through its sequence complementation and plays a role in posttranscriptional gene regulation in different species [19]. Hundreds of different miRNAs found in humans are still conserved in other species, and these conserved miRNAs preferentially interact with human miRNAs. Thus, miRNAs affect almost all human developmental processes, health, and diseases. Indeed, miRNA loss-of-function studies have revealed multiple phenotypes in the brain, eye, muscle, heart, lung, kidney, vascular system, gut, breast, ovary, testis, placenta, thymus, and hematopoietic lineage, as manifested by obvious developmental and cellular physiological defects and behavior abnormality. Many of these developmental and physiological defects affect embryonic or postnatal viability, leading to the other serious conditions, e.g., epilepsy, deafness, retinal degeneration, infertility, immune disorders, or even cancer [20]. To date, hundreds of different miRNAs have been identified in humans, along with thousands of human genes shown as cognate targets of miRNAs [21]. Thus, multiple signaling pathways are involved in the regulation of miRNA [22]. Interestingly, the first miRNA was discovered in *C. elegans,* when molecular geneticists studied the lin-4 and let-7 genes required for nematode development and also found a great surprise that instead of protein-producing mRNA, these genes produced short noncoding mRNAs of 22 nt in length [23,24]. Both lin-4 and let-7 miRNAs can form incomplete complementary pairing with their conserved sites within the 3′-UTR of target gene products, thereby mediating translational repression. Since then, a bulk of miRNAs have also been discovered between humans and mammals, consistent with their transient nature observed in *C. elegans* [25]. Subsequent studies once again confirmed that these endogenous small RNAs are mostly processed by their RNA precursors, each with a hairpin structure, called small RNA because of their small molecular weights [26,27,28]. When compared with prokaryotes, the miRNA-generating pathway in mammals is derived from a more fundamental RNA silencing pathway, namely, RNA interference (RNAi). The miRNA differs significantly from RNAi because the former has evolved as a specific short hairpin structure, which enables its silencing mechanism to target specific mRNAs. By contrast, siRNA does not form a stem-loop structure when it arises from its dsRNA precursors [29,30].

Canonical miRNA processing requires four processes, which begin with synthesis of long-chain RNAs (pri-miRNAs) by functioning of RNA polymerase II [31]. Each pri-miRNA has at least one region that can be self-folded to form the functional substrate of a microprocessor, a heterotrimeric complex containing one molecule of Drosha endonucleases and two DGCR8 chaperones [32]. The pri-miRNAs are processed into pre-miRNAs by the microprocessor and then exported from the nucleus into the cytoplasm through the action of Exportin 5 and RAN-GTP, where they are further processed into miRNAs* by Dicer (an endonuclease with two RNase III domains), as with Drosha [33]. Finally, the formed miRNAs* are loaded into the Argonaute protein with the help of chaperones HSC70/HSP90 to be cleaved into mature miRNAs, and they form a silencing complex, thereby establishing their function [34,35].

Two modes of miRNA functioning in eukaryotic cells are proposed. On the one hand, they form 3′-terminal broad pairing via the Argonaute protein in the silencing complex and utilize its retained endonuclease properties to cleave mRNA in animal cells [36,37]. On the other hand, they form a translation barrier in human cells. This process does not require extensive complementation of miRNAs with target genes, but requires an adaptor protein, TNRC6, which has three homologous proteins in mammals, named TNRC6A/B/C, and is recruited by Argonaute proteins [38]. Then, this complex interacts with the poly(A) tail-associated region (PABPC) of mRNA [39]. Meanwhile, the deadenylase complex and others are also recruited to the 3′-UTR, among which the PAN2–PAN3 complex and the CCR4–NOT complex can cause rapid shortening of the poly(A) tail, resulting in mRNA destabilization, as well as accelerate mRNA uncoating and degradation [40,41]. Furthermore, DDX6, as a helicase, is recruited to inhibit translation by the CCR4–NOT complex, because it interacts with the eukaryotic translation initiation factor 4E (eIF4E) transporter (4E–T), enhances miRNA target attenuation, and inhibits translation initiation [42,43]. The canonical miRNA processing and regulatory mechanisms are schematically represented in Figure 1 [20].

## 3. EMT

In 1967, Betty Hay first proposed the concept of epithelial–mesenchymal transition and described its crucial role in gastrulation and embryonic differentiation. For many years, EMT was predominantly described during embryonic development [44]. During embryonic transformation by EMT, the mesenchyme refers to the nonepithelial cells with so-called amoeba characteristics that exist between the epithelial layers of the embryo, while the epithelium is defined as a tissue in which the mesenchyme is polarized; the front end of the migrating cells differs from the back end due to the lack of organelles. This is reflected by the fact that the Golgi apparatus and centrioles usually occupy the posterior side of migrating cells, while filopodia are most prominently present at the antegrade side of the contact with the basement membrane [45]. With the deepening of research, EMT has mainly been divided into three types according to the biological background of occurrence shown in Figure 2. Type I EMT occurs during embryonic development, whereas type II and III EMT occurs during wound healing, tissue regeneration, and cancer progression. Although a consistent interpretation cannot be provided for the intrinsic diverse and significant differences of the three types of EMT, the interaction of cells and the extracellular matrix is remodeled during this process, enabling epithelial cells to separate from each other and from the underlying basement membrane. After which a new transcriptional program is activated, causing those cells to progressively lose epithelial morphology in favor of mesenchymal morphology [46].

A typical example is the inhibition of E-cadherin expression during EMT, resulting in the typical polygonal, cobblestone-like shape of epithelial cells. Cells acquire spindle-shaped mesenchymal morphology and express markers associated with a mesenchymal cell state, particularly N-cadherin, vimentin, and fibronectin [46]. During this process, those cell junctions, including tight junctions, adherens junctions, desmosomes, and gap junctions are gradually disintegrated, along with a loss of cell polarity, increased actin expression supporting the formation of pseudopodia to facilitate cell adhesion, and enhanced expression of N-cadherin and integrins, but reduced intercellular adhesion in epithelial cells. Furthermore, matrix metallopeptidase family proteins (MMPs) begin to be expressed, thereby degrading the matrix and enhancing the ability of cell invasion. A further diagrammatic representation of cell plasticity during such an EMT transformation is depicted in Figure 3 [15].

## 4. The EMT-Related Transcription Factors

As mentioned above, epithelial–mesenchymal transition not only plays a role in embryonic development and cell differentiation, but also endows cancer cells with higher metastatic and invasive abilities than those of normal cells, which makes the EMT process accompany the occurrence and development of malignant tumors. During EMT, epithelial cells lose their adhesion and tight junctions, leading to a loss of contact with neighboring cells, and hence, resulting in breaks throughout the basement membrane for migration over long distances due to profound changes in the cytoskeletal structure [47]. Most previous studies have shown that classical EMT is accomplished by its relevant transcription factors (EMT-TFs), including the Snail, Twist, and ZEB families. Activation of EMT-TFs increases motility for collective migration of cancer cells in the population, thereby facilitating invasion and dissemination [48]. In addition to activating canonical EMT-related properties, EMT exerts other functions in cancer biology that derive from the multiple additive effects conferred by EMT-TFs.

This is concretely exemplified by the fact that those factors maintain stem-cell properties and even enhance carcinogenicity when associated with cancer stem cells (CSCs), in addition to their involvement in double-stranded DNA repair, evasion of senescence, and induction of anti-apoptosis and pro-survival behavior [49]. Durative adaptation of cancer cells to changing microenvironments enhances tumor cell plasticity in the complex process from its primary locus to remote metastasis. A plausible example is represented by a reciprocal feedback loop between EMT-related transcription factor ZEB1 and the miR-200 family members required for cell differentiation [50]. Considering complex regulatory relationships, the role of EMT-TFs in cancer is also more complicated. Evidence has shown that Snail induces metastasis in breast cancer, while ZEB1 tends to induce pancreatic cancer metastasis. Furthermore, ZEB1 promotes tumor growth, whereas ZEB2 reduces tumor aggressiveness in melanomas. Such tissue specificity and variability of efficacy are other reasons why the functional effects of EMT in cancer biology are often controversial [51,52,53,54]. The extant evidence has revealed that miRNAs control EMT, stemness, and even chemotherapy resistance. Notably, the EMT phenotype is controlled by different miRNAs regulating expression of the same EMT-TFs. Correspondingly, the same miRNA can also produce the same phenotype by controlling the expression of different EMT-TFs, which is a good explanation for the multitarget effects of miRNAs. The expression of epithelial marker genes and the activation of mesenchymal marker genes are mainly accomplished by Snail, Twist, and ZEB during the EMT. Their expression is also shown to be activated in the early stage of EMT, and it plays a central and irreplaceable role in tumor development and fibrosis. Nevertheless, because these transcription factors have distinct expression profiles, their contribution to EMT depends on the cell or tissue types involved and the signaling pathways that initiate EMT. They usually control each other’s expression and cooperate functionally on co-target genes [55]. 

Changed expression of miRNAs may also mediate chemoresistance by acting on other processes, e.g., autophagy or metabolism in hepatocellular carcinoma (HCC) [56]. Co-suppression of Snail1 complexes with SMAD3 and SMAD4 in mammary epithelial cells results in TGF-β-mediated repression of E-cadherin and Occludin expression [57]. Multiple signaling pathways can facilitate EMT by activating the expression of Snail1. For example, in MDCK cells, Snail1 cooperates with ETS proto-oncogene 1 (ETS1) and Sp1 transcription factor (Sp-1) to activate matrix metallopeptidase family (MMP) expression by activating mitogen-activated protein kinase (MAPK) and phosphatidylinositol-4,5-bisphosphate 3-kinase (PI3K) [58]. Glycogen synthase kinase 3 beta (GSK-3β) enables successive phosphorylation of the Snail protein to control its abundance, further monitoring EMT in cancer cells [59]. Although EMT-TFs play a role in different tissue-derived cancers, a completely direct link between them has not been presented. This is due to the fact that deletion of Snail1 and Twist1 genes in pancreatic cancer did not directly attenuate metastasis, but conversely increased chemoresistance in mouse transgenic models [51]. In addition, the basic helix–loop–helix (bHLH) family of transcription factors are also important regulators of development and differentiation. They bind to target DNAs through homo- or heterodimer formation at a consensus E-box site to regulate gene transcription, among which E12 and E47, Twist1 and Twist2, and inhibitor of DNA-binding proteins (IDP) each play a key role in the EMT progression [55]. This process has also been shown to be regulated by newly discovered transcription factors, such as forkhead box (FOX) transcription factors, which are defined by their DNA-binding forkhead domains [60]. Taken together, EMT shapes the cellular physiological behavior at multiple levels by integrating epigenetic modification, transcriptional control, alternative splicing, protein stability, and subcellular localization. The mechanism for triggering EMT is not simply linear, but should be systematic, multidimensional, and holistic, within multi-hierarchical complex networks [15]. As shown in Figure 4, different transcription factors play distinctive transcriptional regulatory roles in the EMT process and crosstalk with one another.

## 5. miRNAs Control EMT via the EMT-Related Transcription Factors

As aforementioned, the emergence and development of EMT are caused by a systemic change in cells. All EMT-TFs coordinate the repression of epithelial genes with the induction of mesenchymal genes, as accompanied by fundamental changes in the cellular regulatory network. Herein, we emphasize that the same transcription factors can perform functions at different levels, including transcription and translation, expression of noncoding RNAs, alternative splicing, and repression and activation of protein stabilization [61].

### 5.1. miRNAs Regulate Tumor Cell EMT through the Transcription Factor Snail1/2

Snail1/2 is one of the main regulators of EMT with a clear correlation between both. Snail1 is targeted by miR-29b and miR-30a, as EMT inhibitors, to repress expression of E-cadherin, thereby reversing the EMT process in prostate cancer and losing its migration and invasion ability [62]. Low expression of miR-30c promotes renal tissue fibrosis, whereas high expression of miR-30c targeting Snail1 downregulates the process of fibrosis, as well as blocks EMT, thus protecting kidney function from damage caused by diabetes in diabetes-induced renal cancer [63]. Both miR-137 and miR-34a were reported in ovarian cancer to directly target Snail1 to regulate EMT in ovarian cancer cells so as to enhance invasiveness and spheroid formation properties [64]. The expression of Snail1 targeted by the miR-15 family forms a regulatory feedback loop, thereby regulating the process of cellular EMT [65]. Activation of p53 relieves repression of the induced transcription factor Snail1 by downregulating miR-34a/b/c genes, leading to enhanced migration and invasion. Meanwhile, miR-34a also downregulated Snail2 and ZEB1, as well as stem-cell-specific factors BMI1, CD44, CD133, olfactomedin 4 (OLFM4), and c-Myc. Intriguingly, the transcription factors Snail and ZEB1 can also bind to the E-box of the miR-34a/b/c promoter, conversely inhibiting the expression of miR-34, miR-204a, and miR-34b/c [66]. Similarly, the mechanism via which miR-34 targets Snail was also demonstrated in colon cancer cells with p53 dysfunction [67]. To impair the occurrence of EMT in prostate cancer, a double negative feedback loop mechanism was structured among miR-1, miR-200b, and Snail2 [68]. High expression of miR-204/211 facilitates the shaping of retinal epithelial cells. Studies have found that miR-204/211 targets Snail2 to regulate the expression of cadherin and increase the plasticity of myoepithelial cells [69]. Both miR-182 and miR-203 regulate CDH3 by targeting Snail2, resulting in an EMT phenotype [70,71]. In gastric cancer cells, miR-33a targets the transcription factor Snail2 in regulating cell invasion and migration [72]. The expression of Snail2 is inhibited by miR-506 binding to the 3′-UTR of Snail2, leading to increased E-cadherin expression and inhibited cell proliferation and migration in gastric cancer [73]. miR-124 targets Snail2 and inhibits EMT in prostate cancer [74]. Similarly, miR-124 regulates differentiation and migration by regulating Snail2 and IQGAP1 during embryonic stem cell development to ensure their stemness [75].

### 5.2. miRNA Regulates the Process of Cancer EMT via the Transcription Factor ZEB1/2

As an indispensable EMT transcriptional repressor, the ZEB family plays multifarious biological functions in organisms. For example, it promotes cell proliferation and migration, as well as triggers drug resistance and immune mechanisms. Conversely, inhibition of ZEB expression upregulates epithelial gene expression and causes EMT blockage [76]. Almost all of the miR-200 family members (i.e., miR-200a, miR-200b, miR-200c, miR-141, and miR-429), along with miR-205, were found to be significantly downregulated in breast cancer cells. The obtained results demonstrate that overexpression of the miR-200 family leads to inhibition of TGF-β-induced EMT, which serves as a target of transcriptional repressors ZEB1 and ZEB2 that activate the expression of E-cadherin [77]. miR-200c targets ZEB1 to control the proliferation and invasion of tumor stem cells. Overexpression of miR-200c significantly downregulates the expression of ZEB1 and vimentin, but upregulates E-cadherin expression, resulting in decreased colony formation, migration, and invasion, as well as a reduction in metastatic tumor cells in mouse lung xenografts [78]. Expression of p53-activated miR-200 and miR-192 (a homolog of miR-215) targeted ZEB1/2 to restrain EMT in hepatoma cells [79]. Furthermore, miR-200f has also been shown to form a molecular junction between Snail and ZEB to participate in the EMT regulation [80]. Of note, clinical studies have revealed that the low expression of miRNA-199b-3p in ovarian cancer is negatively correlated with the key regulators of EMT, whereas the high expression of miRNA-199b-3p has a longer overall survival. miRNA-199b-3p can target and regulate ZEB1 so as to induce the expression of E-cadherin and organize the transition of epithelial cells to mesenchymal cells [81]. In addition, miR-128-3P targets ZEB1 to hinder invasion and migration in pancreatic cancer cells [82].

### 5.3. miRNA Regulates Tumor Cell EMT via the Transcription Factor Twist

It is generally accepted that the Twist family encodes transcription factors containing basic helix–loop–helix structures (bHLHs), which form functional homodimers and heterodimers in the nucleus, bind consensus DNA E-box sequences, and promote tumor cell invasion and metastasis. Studies have revealed that miR-495 inhibits the expression of Twist1 and inhibits the proliferation and metastasis of gastric cancer cells [83]. Both miR-15a-3p and miR-16-1-3p target two conserved sites in the 3′-UTR of Twist1 to reduce the mRNA and protein levels of biomarkers, including N-cadherin, *α*-SMA, fibronectin, and MMP9, so as to attenuate the migration and invasion of gastric cancer cells [84]. The competitive binding of miR-129-5p to Twist reverses the EMT process of drug-resistant breast cancer cells caused by long noncoding RNAs (lncRNAs), thus improving cell drug sensitivity and slowing migration and invasion [85]. miR-381-3p inhibits breast cancer progression and EMT progression by targeting Sox4 and Twist1 expression and downregulating Smad protein, which is downstream of TGF-β [86]. miR-9-5p activates N-cadherin by acting on Twist1 to promote EMT in cervical squamous cells, and promotes EMT in adenocarcinoma cells by inhibiting cadherin 1 (CDH1) and activating cadherin 2 (CDH2) [87]. miR-98 targets Twist to activate E-cadherin expression to reduce the invasion and metastasis of non-small-cell lung cancer. In addition, overexpression of miR-98 activates the Akt/bcl2/Bax signaling axis to induce apoptosis and improve patient prognostic indicators in non-small-cell lung cancer (NSCLC) [88]. Therefore, miR-186 targets Twist1 to inhibit the expression of N-cadherin, vimentin, and matrix metallopeptidase 9 (MMP9), as well as increase the expression abundance of E-cadherin to inhibit the proliferation, migration, invasion, and EMT of cholangiocarcinoma cells [89]. miR-203 can target Twist1 to inhibit cell migration and viability, while inhibition of miR-203 and simultaneous interference with Twist1 can promote EMT in bladder cancer cells [90]. Previous clinical data showed that the expression of miR-300 is decreased in head and neck squamous cell carcinoma (HNSCC) patient specimens. The study found that miR-300 directly targets the 3′-UTR of Twist, and overexpression of miR-300 blocks the invasion of cancer cells in vitro and metastasis of mouse lung tumor nodules in vivo [91]. miR-539 targeting Twist1 inhibited the growth of pancreatic cancer xenografts in nude mice, as well as attenuated the migration and invasion of pancreatic cancer cells in vitro [92]. miR-361-5p suppresses cell proliferation and invasion, and reverses EMT by targeting Twist in liver cancer cells [93]. In liver cancer cells that secrete alpha-fetoprotein (AFP), miR-675 targets Twist1 and Rb1 to activate the expression of the epithelial marker gene *CDH1* and inhibits the expression of the mesenchymal marker gene *Vim* to reduce the invasive ability of HCC and block EMT [94]. Thus, it is reasoned that such differences obtained from previous results are attributable to the fact that distinct effects of miRNAs were exerted in diverse tumor microenvironments.

### 5.4. miRNAs Regulate Tumor Cell EMT through Other Transcription Factors

It has been determined that miR-221/222 targets transcriptional repressor GATA binding 1 (TRPS1), a direct repressor of the ZEB2 expression to regulate EMT. Thus, overexpression of miR-221/222 is beneficial for reducing the TRPS1 abundance, alleviating the inhibitory effect of ZEB2, and promoting the expression of interstitial genes, thereby accelerating the migration and invasion of breast cancer cells [95]. It has been reported that miR-145 interacts with the transcription factor Oct4 and mediates β-catenin overexpression of the transcription factors Snail and ZEB1/2, thus affecting the EMT process of breast cancer cells [96]. Sterol regulatory element-binding transcription protein 1 (SREBP1), which is highly expressed in breast cancer cells, was reduced by targeting of miR-18a-5p. Low expression of SREBP1 reduces the deacetylated inhibition of E-cadherin, attenuates the effect of the Snail/HDAC1/2 inhibitory complex, and blocks breast cancer cell growth and metastasis [97]. Prrx1, as a novel EMT-inducing factor, is targeted and inhibited by miR-655 in breast cancer cells, so as to halt cell migration and invasion during tumor EMT progression [14]. Interestingly, it was found that the transcription factors Snail1 and Prrx1 are expressed in a complementary manner in vertebrate development and carcinogenesis, in which Snail1 directly represses Prrx1 expression but, in turn, is directly activated by Prrx1. miR-365 can interact with TTF-1 or NK2 homeobox 1 (NKX2-1) and the 3′-UTR of high mobility group AT-hook (HMGA2) to inhibit EMT in lung cancer cells [13]. miR-451 directly targets the transcription factor c-Myc to inhibit the invasion and metastasis of docetaxel-resistant lung adenocarcinoma (LAD) cells in vitro and in vivo. A high abundance of c-Myc induces glycogen synthase kinase 3β-dependent extracellular signal-regulated kinase (ERK) inactivation and Snail activation [98]. Of note, the virion can act on the tumor suppressor PTEN with the help of host-expressed miRNA to regulate the PI3K/Akt/GSK-3β signaling pathway, which ultimately leads to the high expression and nuclear aggregation of Snail and β-catenin, resulting in an inhibition of epithelial gene expression and a promotion of interstitial gene expression to facilitate EMT [99]. miR-19 inhibits epithelial gene expression and enhances mesenchymal gene expression by targeting PTEN to regulate EMT in lung cancer cells [6]. The same on-target effect has also been demonstrated under exosome-mediated effects [100]. miR-301a targets the tumor-promoting factor WNT1 to increase the radiosensitivity of esophageal squamous cell carcinoma (ESCC) cells, while suppressing drug resistance and EMT by targeting Snail and Vimentin [11]. The regulation of EMT by all these transcription factors described in this review and their references are listed in Table 1.

## 6. Non-Transcription Factors Are Targeted by miRNAs to Shape EMT in Different Tumors

It is evident that miRNAs can target non-transcription factor proteins that contribute to defining epithelial or mesenchymal phenotypes. They include those encoding adhesion junction and polarity complex proteins, signaling mediators, and those controlling the expression of the EMT transcription factors listed in Table 1.

### 6.1. miRNAs Regulate EMT in Breast Cancer

Breast cancer is a malignant tumor that occurs in the mammary epithelium or ductal epithelium. Previous studies have found that miRNAs can regulate the EMT process of breast cancer cells by controlling EMT marker proteins or other non-transcription factor targets. For example, miR-9 targets CDH1, and overexpression of miR-9 inhibits the expression of E-cadherin, promotes the mesenchymal phenotype, and increases cell migration and invasion [12]. miR-31 decelerates breast cancer metastasis by targeting genes such as integrin *α*5 (ITGA5), radixin (RDX), and Ras homolog family member A (RhoA) [101]. Likely in breast cancer, bone morphogenetic protein 6 (BMP6) competitively binds to the miR-21 promoter through the E2-box- and AP-1-binding sites, leading to inhibition of the expression of miR-21 and enhancing E-cadherin to achieve EMT inhibition [102]. The miR-200 family can regulate the metastasis of breast cancer cells by controlling abundance of serine/threonine kinase (AKT) protein isoforms [103]. miR-661 targets the mRNA 3′-UTR of the cell adhesion protein Nectin-1 and lipid transferase StarD10 to reduce their stability, while abnormal expression of the latter relieves the inhibition of Snail1 and indirectly regulates the metastasis of breast cancer cells [104]. miR-146a targets thioredoxin-interacting protein (TXNIP) to promote breast cancer cell fibrosis and exacerbate cell metastasis in the EMT process [105]. miR-155 promotes this EMT process of breast cancer cells by targeting RhoA to disrupt cell tight junctions by TGF-β induction, enhancing cell migration and invasion [8].

### 6.2. The Regulatory Role of miRNA in Gastric Cancer EMT

Gastric cancer is one of the most morbid cancers worldwide, with a 5 year relative survival of approximately 20%. miR-2392 targets the regulatory operator-like protein 3 (MAML3) and NSD histone methyltransferase 1 (WHSC1) to inhibit the expression of the transcription factors Slug/Twist1 and block gastric cancer cell metastasis in vivo and in vitro [106]. miR-4521 inhibited by the ETS proto-oncogene 1 (ETS1) targets both insulin = like growth factor 2 (IGF2) and forkhead box M1 (FOXM1) to inhibit Akt/GSK3β/Snai1 pathway proteins, and achieves inhibition of gastric cancer cell metastasis in hypoxia-induced conditions [107]. miR-214 was found to target fibroblast growth factor 9 (FGF9) in gastric cancer cells, and an experiment showed that it prevented cell fibrosis, increased the expression of E-cadherin, and decreased the expression of vimentin and N-cadherin to negatively regulate EMT [108]. miR-194 targets the N-cadherin expression to reduce the metastasis and invasion of gastric cancer cells [109].

### 6.3. miRNAs Regulate EMT in Liver Cancer

Hepatocellular carcinoma is a malignant tumor derived from hepatocytes and hepatobiliary cells. The latest data show that liver cancer ranks sixth in new cases and third in mortality worldwide [110]. Studies found that a low expression of miR-124 promotes cell migration, but overexpression of miR-124 can directly target Rho-associated coiled-coil containing protein kinase 2 (ROCK2) and enhancer of Zeste 2 polycomb repressor complex 2 subunit (EZH2) genes to disrupt the stability of their mRNAs and inhibit EMT [111]. In hepatoma cells, miR-205 promotes apoptosis and inhibits proliferation by targeting axon guidance factor 4C (SEMA4C), as well as attenuates tumor cell EMT [112]. In contrast, the expression of miR-194 in hepatic stromal cells resulted in decreased N-cadherin levels, preventing cell migration and invasion [17]. miR-532-3p regulates HCC cell metastasis and invasion through Gankyrin-dependent activation of Twist by targeting the tumor-promoting factor kinesin family member C1 (KIFC1) [113]. Low expression of miR-34 leads to poor prognosis of liver cancer patients and enhanced migration and invasion of hepatocytes in vitro. Studies have previously found that miR-345 indirectly regulates the transcription factors Slug, Snail, and Twist by targeting interferon regulatory factor 1 (IRF1) to activate the mTOR/STAT3/Akt signaling pathway, which can slow down the EMT process of hepatoma [114]. miR-148a reduces the accumulation of Snail by binding to the tyrosine kinase receptor factor (Met) for hepatocyte growth, which can activate the phosphorylation of downstream Akt Ser473 and inhibit the phosphorylation of GSK-3β-Ser9, leading to relief of EMT in liver cancer [115].

### 6.4. miRNAs Regulate EMT in Renal Cancer 

Renal cancer is a malignant tumor caused by the canceration of epithelial cells in different parts of the renal parenchyma of the urinary tubule. miRNAs regulate renal cancer cell migration by preventing cellular fibrosis. There is evidence that miR-214-3p can improve fibrosis and reverse EMT in renal cancer cells by targeting E-cadherin under hypoxia-inducible conditions [116]. miR-328 directly regulates CD44 to suppress stress-induced renal tubular epithelial cell fibrosis and alleviate EMT [117]. Unlike other cell carcinomas, p53-dependent activation of miR-34a targets the fibrosis inhibitor Klotho protein to promote EMT in renal tubular epithelial cells [118]. miR-199b-3p can target KDM6A to regulate E-cadherin expression to prevent EMT, and thus improve renal tissue damage [119]. miR-485-5p, inhibited by the circular RNA circPTCH1, arrests the invasion and metastasis of renal cell carcinoma by targeting the downstream MMP14 [120].

### 6.5. miRNAs Regulate EMT in Ovarian Cancer 

Ovarian cancer is a common malignant tumor of female reproductive organs, and its incidence is second only to cervical cancer and uterine cancer. According to the literature, miR-222-3p binds to the 3′-UTR of programmed cell death 10 (PDCD10) and inhibits the translational expression of PDCD10, which induces EMT by downregulating E-cadherin and enhancing vimentin. Conversely, overexpression of miR-222 decreased the abundance of PDCD10 and inhibited ovarian cancer epithelial cell migration in vitro and in vivo [121]. miR-183-5p targets the Ezrin protein to promote apoptosis of human endometrial cancer cells and inhibit EMT, proliferation, invasion, and migration [122].

### 6.6. miRNAs Regulate EMT in Pancreatic Cancer 

Pancreatic cancer is a malignant tumor arising from the exocrine pancreas. miR-144, as a member of the miR-200 family, regulates the colocalization of Snail by targeting the cell surface receptor Notch-1 protein-activated microtubule-associated protein-1 (DCAMKL-1) to regulate EMT in pancreatic cancer cells [123]. miR-3656 targets RHOF, a member of the Rho subfamily of small GTPases, so as to resist the resistance effect of adenocarcinoma to chemotherapeutic drugs, thereby preventing the growth, invasion and migration of pancreatic cancer cells [124]. On the basis of the abovementioned mechanism, a new targeted nano-delivery system using liposomes to prepare siDCAMKL-1 nanoparticles for targeted therapy has been well validated in nude mouse xenograft experiments [125].

### 6.7. miRNAs Regulate EMT in Rectal Cancer 

Colorectal cancer is a malignant tumor originating from the glandular epithelium of the colorectum. miR-130b targets peroxisome proliferator-activated receptor *γ* (PPARγ) to regulate the invasion and metastasis of colorectal cancer cells [126]. miR-146a targets Numb to stabilize β-catenin in spherical colon cancer stem cells, while Snail induces the expression of microRNA-146a through the β-catenin–TCF4 complex to form a stable feedback loop to maintain the cell stemness and prevent EMT [127]. miR-17-5p inhibits colorectal cancer metastasis in mice and in vitro by directly binding to the 3′-UTR of vimentin mRNA [5].

### 6.8. miRNAs Regulate EMT in Oral Cancer 

Oral cancer is a general term for malignant tumors that occur in the oral cavity. miR-34a-5 inhibits AXL activation of the Akt/GSK-3β/β-catenin signaling pathway to promote the nuclear translocation of β-catenin, which activates Snail to transcriptionally upregulate both MMP-2 and MMP-9 expression and increases invasiveness and mobility in oral squamous carcinoma cells (OSCC) [128]. miR-3187-3p enhanced cell migration and invasion by activating the Wnt/β-catenin signaling pathway and targeting PER2 in cervical squamous cell carcinoma [129]. Additionally, miR-10b targets and regulates E-cadherin to promote laryngeal cancer metastasis [16].

### 6.9. miRNAs Regulate EMT in Lung Cancer 

Lung cancer refers to malignant tumors originating from the trachea, bronchi, and lungs, which include several major types of squamous cell carcinoma, adenocarcinoma, small-cell carcinoma, and large-cell carcinoma. miR-145 and miR-497 negatively regulate the EMT through co-inhibiting the TGF-β signaling pathway and targeting metadherin (MTDH) in NSCLC [130]. Interestingly, clinical studies have shown that the methylation of miRNA-34 and miR-200 promoters can affect the EMT transformation of tumor cells to a certain extent, and further studies have found that high expression of miRNA-34 and miR-200 has a significant effect on patients [131]. A targeted linkage of miR-138 to both G-protein-coupled receptor kinase-interacting protein 1 (GIT1) and SEMA4C inhibits EMT in NSCLC [132]. miR-483-5p targets the Rho protein guanylate dissociation inhibitor (RhoGDI1) and activates the leukocyte adhesion molecule (ALCAM). Only when ALCAM, as an EMT inhibitor, is dysfunctional in lung adenocarcinoma cells is the EMT program also activated [133].

### 6.10. miRNAs Regulate EMT in Other Cancers 

Wnt3a, a tumor-promoting factor, negatively regulates the expression of miR-497 through c-Jun. Overexpression of miR-497 effectively inhibited the expression of Wnt3a, which increased the expression of E-cadherin, but also reduced the expression of N-cadherin and vimentin. Ultimately, it slowed the proliferation and metastasis of glioma cells and alleviated the cellular EMT process [134]. Interestingly, circular RNA (circRTN4) interacted with miR-497-5P to regulate the downstream RAB11FIP1 protein to block the ubiquitination, which inhibited Snail1/2, Twist, ZEB1, and N-cadherin to promote the metastasis of pancreatic ductal carcinoma cells [135]. miR-139-5p regulates the transcription factor Twist and the mesenchymal marker proteins N-cadherin and vimentin by targeting the sex-determining region Y-box protein 5 (SOX5) to achieve EMT inhibition [136]. In the activated state of Nrf2, miR-29a/b targets desmocollin-2 and impairs keratinocyte hyper-adhesive desmosome formation [137]. miR-448 targets a specific AT-rich sequence-binding protein 1 (SATB1) to activate the MAPK signaling pathway, while enhancing Twist promotes EMT [138]. Notably, miRNA-targeted regulation of EMT does not entirely promote cell metastasis and invasion; for instance, miR-100 downregulates E-cadherin by targeting SMARCA5, a regulator of *CDDH**1* promoter methylation, but also targets other oncogenes. HOXA1 inhibits tumor cell initiation and motility in vitro and in vivo; its biological performance shows the opposite trend to the conventional EMT [139]. Functioning of miR-24 depends on the TGF-β/SMAD and MAPK/ERK signaling pathways to target neuroepithelial transforming factor 1A (NET1A), an Rho-GEF, which activates RhoA to disrupt tight junctions between cells and promote EMT [140]. miR-1236 targets histone deacetylase 3 (HDAC3) and SUMO-specific protease 1 (SENP1) to slow down cellular EMT, while Twist1 inhibition of miR-1236 under hypoxic conditions relieves the inhibition of HDAC3 and SENP1; thus, stabilizing HIF-1α continuously activates Twist1 and forms a regulatory feedback loop network [141]. In salivary gland tumor tissue samples, miR-155, miR-200c, miR-9, miR-138, and miR-200c were differentially expressed in pleomorphic adenoma (PA), mucoepidermoid carcinoma (MEC), and adenoid cystic carcinoma (ACC). The obtained results show that the EMT transformation of salivary gland tumors is controlled by different transcriptional regulators [142]. These miRNAs targeting non-transcription factors to modulate EMT in major cancers are listed in Table 2.

## 7. Perspective

As emphasized herein, the biological functions of miRNAs are complex, and tumor cells with systematic changes often exhibit specific and systematic regulation. The exploration of EMT should be combined with clinical trials, animal models, and model cell lines from different types of tumors. Further endeavors need to be focused on clinical development, especially in the areas of drug resistance and tumor targeted therapy for miRNA families, such as miR-200 and miR-34, whose biological functions have been well documented in EMT. Those emerging miRNAs that are not well understood need to be continuously explored with clinical data and bioinformatic analysis to further analyze their biological functions. Undoubtedly, these processes remain to be refined and differentiated, which is primarily embodied in the diversification of cancer types and targets. An urgent assignment to screen tumor markers warrants to be solved within distinct contexts of EMT, based on different tumor microenvironments. Whether aberrantly expressed miRNAs can be regarded as indicators of tumorigenesis and risk assessment also requires in-depth studies and further demonstrations. Certainly, with the in-depth study of miRNA regulation of EMT, more information will be gradually unveiled, opening up a new strategic view for tumor prevention and treatment.

## Figures and Tables

**Figure 1 cells-11-01981-f001:**
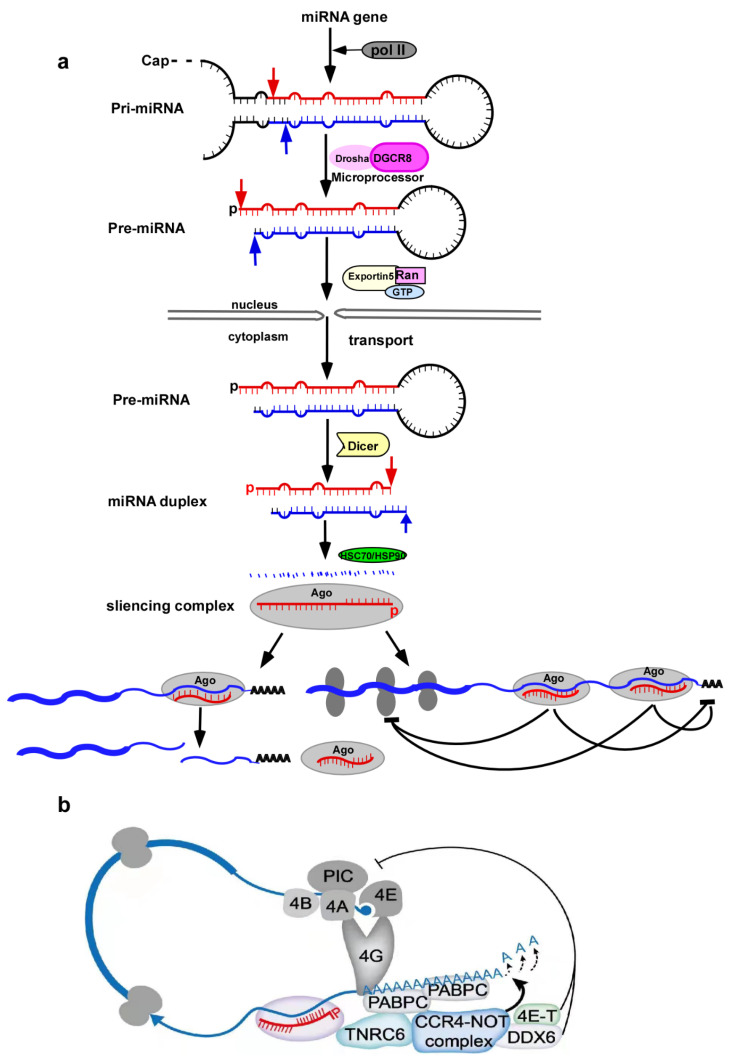
Canonical miRNA processing and relevant mechanism of silencing complex. (**a**) Located in the nucleus, miRNA precursors with cap-ring and hairpin structures are transcribed by the action of RNA polymerase II, transported into the cytoplasm, and loaded into AGO proteins to form silencing complexes, before exerting their biological roles. (**b**) The silencing complex recruits a number of complex proteins, including TNRC6, CCR4–NOT, and DDX6. The silencing complex binds the 3′-UTR of mRNA, thereby accelerating mRNA degradation and inhibiting translation initiation.

**Figure 2 cells-11-01981-f002:**
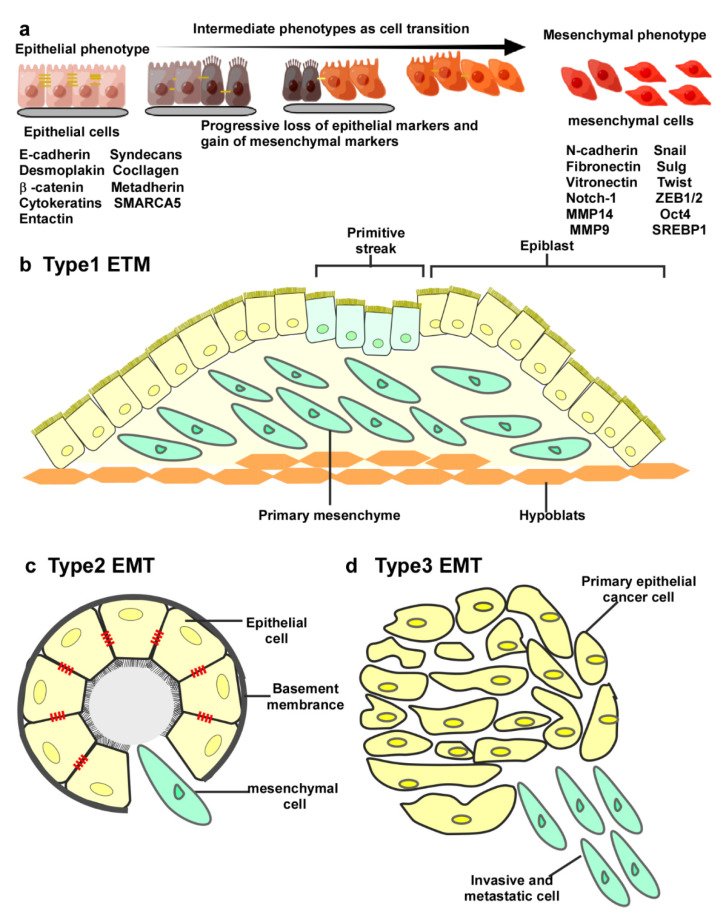
EMT transfer schematic diagram and three typical categories. (**a**) During the transition from an epithelial phenotype to mesenchymal phenotype, cells lose their intercellular connections and gradually separate from the basement membrane. Four mesenchymal genes, including interstitial N-cadherin and fibronectin, are upregulated, whereas E-cadherin, vitronectin, desmoplakin, and laninmin are downregulated. (**b**) Type I EMT is associated with embryo implantation and gastrulation. The ectoderm generates primary mesenchyme through EMT and then forms the secondary epithelium and gradually differentiates into other cells. (**c**) Type II EMT is associated with wound healing, is more persistent than type I, and occurs primarily in the context of inflammation. (**d**) Type III EMT is associated with tumor metastasis and facilitates tumor metastasis and invasion.

**Figure 3 cells-11-01981-f003:**
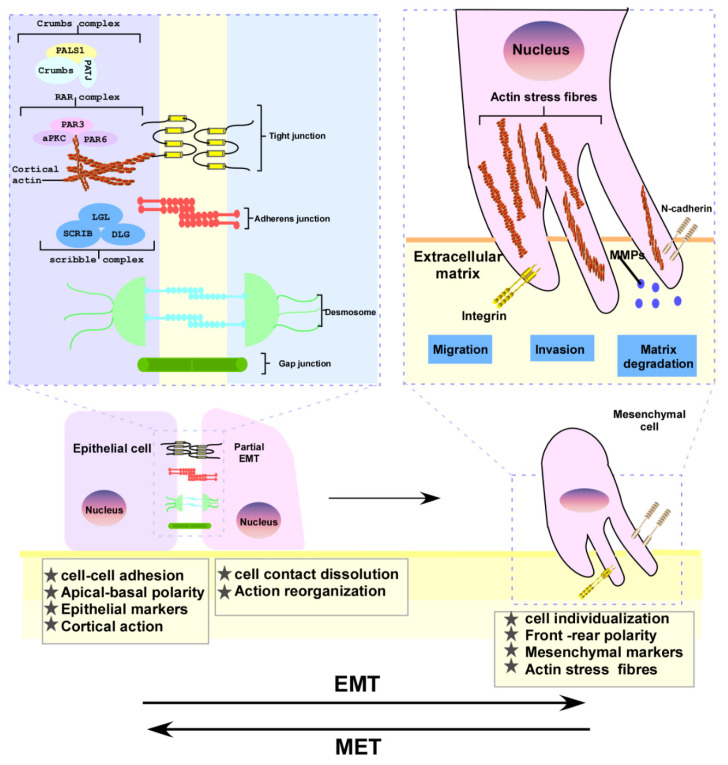
Schematic diagram of EMT plasticity during metastasis of cancer cells. In this process, the tight junctions, adhesions, desmosomes, and gap junctions are gradually dissolved. In the meantime, N-cadherin, MMPs, and integrin are upregulated to promote migration and invasion.

**Figure 4 cells-11-01981-f004:**
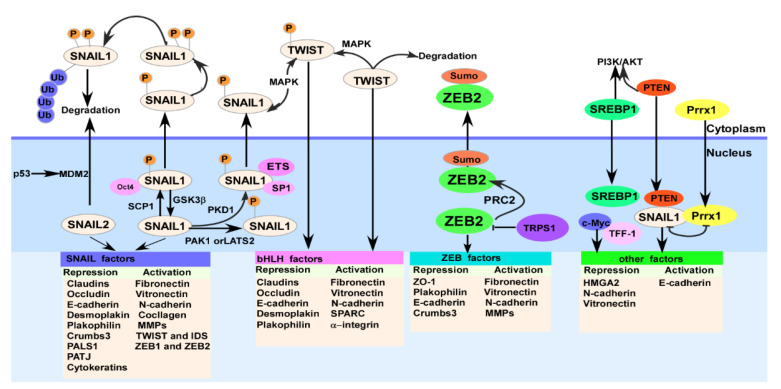
Multiple transcription factors regulate cellular EMT process. These transcription factors include Snail1/2, Twist, and ZEB1/2, whose activation speeds up cellular EMT, while other transcription factors, such as SREBP1, Prrx1, and C-MYC, slow down cellular EMT. Notably, certain interactive and feedback regulatory loops also exist between different transcription factors.

**Table 1 cells-11-01981-t001:** Distinct miRNA-Targeted Transcription Factors to Regulate Tumor EMT.

**miRNA**		EMT-TFs	**Tumor Type**	**Refs.**
miR-29b	miR-34a/b/		prostate cancer	[62]
miR-15		SNAIL1	kidney cancer	[63]
miR-30a/c			ovarian cancer	[64]
miR-34a	miR-1			
miR-33a	miR-200b		breast cancer	[66]
miR-203	miR-204/211	SNAIL2	prostate cancer	[68,72]
miR-182	miR-506		gastric cancer	[73,74]
miR-124				
miR-200a	miR-34a		lung cancer	[78]
miR-200b	miR-192		liver cancer	[79]
miR-200c	miR-200f	ZEB1/2	breast cancer	[77]
miR-141	miR-128-3p		kidney cancer	[80]
miR-429			pancreatic cancer	[82]
miR-199b-3p			ovarian cancer	[81]
miR-361-5p	miR-129-5p		bladder cancer	[90]
miR-9-5p	miR-495		non-small cell lung cancer cancer	[88]
miR-15a-3p	miR-98		liver cancer	[93,94]
miR-186	miR-16-1-3p	TWIST1	lung cancer	[91]
miR-203	miR-300		cervical cancer	[87]
miR-381-3p			gastric cancer	[83,84]
miR-675			cholangiocarcinoma	[89]
miR-539			breast cancer	[85,86]
miR-365		TTF-1	lung cancer	[13]
miR-145		Oct4	breast cancer	[96]
miR-221/222		TRPRS	NCI60	[95]
miR-451		c-Myc	lung adenocarcinoma	[98]
miR-18a-5p		SREBP1	breast cancer	[97]
miR-655		Prrx1	breast cancer	[14]

**Table 2 cells-11-01981-t002:** Distinct miRNAs Targeting Non-Transcription Factors to Regulate Cancer EMT.

Tumor Type	miRNA				Target Gene				Refs.
Breast	miR-661	miR-200	miR-146	miR-31	*CDH1*	*ITGA5*	*RhoA*	*TXNIP*	[8,12]
	miR-155	miR-9	miR-21		*Nectin-1*	*AKT*	*StrD*		[101,102,103,104,105]
Liver	miR-148a	miR-205	miR-124	miR-34	*ROCK2*	*EZH2*	*Met*	*CDH2*	[17]
	miR-532-3p				*SEMA4C*	*IRF1*	*KIFC1*		[110,111,112,113,114,115]
Lung	miR-483-5P	miR-497	miR-145	miR-138	*ALCAM*	*GIT1*	*MTDH*	*SEMA4C*	[130,131,132,133]
Gastric	miR-2392	miR-214	miR-194		*FGF9*	*CDH2*	*IGF2*	*FOXM1*	[106,107,108,109]
	miR-4521				*WHSC1*				
Kidney	miR-214-3p		miR-328	miR-34a	*CD44*	*Klotho*	*MMP14*	*KDM6A*	[116,117,118,119,120]
	miR-485-5p		miR-199b-5p		*CDH1*				
Oral	miR-34a-5p	miR-10b	miR-3187-5p		*CDH1*	*PER2*	*AXL*		[16,128,129]
Recta	miR-130	miR-146a	miR-17-5p		*PPAR* *γ*	*Vim*	*Numb*		[5,126,127]
Ovary	miR-222-3p		miR-183-5p		*PDCD10*	*Ezrin*			[121,122]
Pancreas	miR-144	miR-3656			*Notch-1*	*RHOF*			[123,124,125]

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
