# Peer review of "The Role of MicroRNA in the Regulation of Tumor Epithelial–Mesenchymal Transition"

_cells, 2022, doi:10.3390/cells11131981_

Round 1

Reviewer 1 Report

The article is a review of the latest studies about relationships between microRNA and epithelial to mesenchymal transition in different tumors. After a general description of miRNAs mechanisms, it covers all the most common type of human tumors, explaining the different role of miRNAs as enhancers or suppressors of EMT transcription factors.

The embryonic developmental process mediated by EMT is also well explained, highlighting its crucial role in physiological processes as well.

EMT transcription factors are well described in carcinogenicity and it is well related to the expression and regulation function of different miRNAs.

Even if English language is sometimes not correct, the manuscript is well structured and it is a complete review of most recent publications.

Images and scheme are clear and appropriate.

Overall, the work fit the journal scope and it may be interesting for the readership of the journal even if the work doesn’t represent and advance in the current knowledge.

English language is sometimes not appropriate.

Line 232: the phrase is not clear, maybe it lacks of a verb?

line 333 to line 340 and line 353 to line 355: english is not clear. The concepts are not well expressed as in the original articles because of the incorrect use of English.

line 367: do you mean different type of carcinomas?

Author Response

POINT 1: Line 232: the phrase is not clear, maybe it lacks of a verb?

Response: We are greatly grateful to point out issue. Now, we have rewritten the sentence to make a clear explanation of the change of EMT that is systematic, rather than single linear.

POINT 2: line 333 to line 340 and line 353 to line 355: English is not clear. The concepts are not well expressed as in the original articles because of the incorrect use of English.

Response: We are greatly thanks to this reviewer for the valuable suggestions. Now, we have corrected and re-edited all relevant sentences to give a better understanding of the text for a wide readership. Also, we double checked the original text that microRNA-451 induces the epithelial-mesenchymal transition in docetaxel-resistant lung adenocarcinoma cells by targeting proto-oncogene c-Myc.

POINT 3: line 367: do you mean different type of carcinomas?

Response: We appreciate that this referee raised this question. Our original intention was to express that miRNAs can play a regulatory role in different types of tumours through non-transcription factors. For example, the same miRNA can produce similar but different biological effects in different types of tumour cells. A typical example is the miR-200 family that can regulate both breast cancer and pancreatic cancer. This is mainly reflected by target proteins regulated by miRNA, so that different biological effects will be exerted by different target proteins. Such target proteins do not act as de facto transcription factors, thus serve as non-transcription factors. For this, this subtitle has been restated.

Reviewer 2 Report

In this review, Feng et al describe the miRNA contribution to EMT molecular mechanism. The review describes in 21 paragraphs, miRNA biological function in the context of cancer and metastasis, miRNA biogenesis and their mechanism of action, the EMT process and the main transcription factors involved in the mechanism of action, the role of miRNA in controlling expression of EMT transcription factors like, SNAIL, ZEB, TWIST and others. Finally, the review describes the role of miRNA in the regulation of non-transcription factors that regulate EMT in several model of epithelial cancers.

The review is interesting for the field and might deserve to be published pending extensive revisions.

1- The writing needs to be proofread by a native English speaker, a lot of sentences need to be clarified to make them accessible to the reader. For instance, sentences line 29 or lines 87 to 90 or 235 to 238, are difficult to understand.

2- Authors need to reread the text to correct mistakes leading to confusion or false informations. For instance, the sentence line 103 mentions that the two modes of miRNA biological function operate in the nucleus whereas it occurs in the cytoplasma.

3- The plan of the review needs to be rethinked in order to improve the clarity of the text. An effort of structuration should be made in order to avoid the “catalog effect” of the review.

4- A paragraph placing the expression of key miRNA involved in EMT process in the context of clinical assessment such as prediction of metastasis, survival etc… would be a plus.

5- The citation of previous recent studies and reviews such as Kerche et al 2022 PMID: 34981333, Garinet et al 2021 PMID: 34642464, Wei et al 2021 PMID: 33416170, Ashrafizadeh et al 2020 PMID: 32664703, would be interesting.

6- The miR-199b-3p is an interesting miRNA in the field and deserve also to be cited.

Author Response

Thank you for your carefully reviewing this mini-review. We have paid a great attention to your valuable suggestions on language and relevant contents, all of which have been corrected in this revised version.

POINT 1: The writing needs to be proofread by a native English speaker, a lot of sentences need to be clarified to make them accessible to the reader. For instance, sentences line 29 or lines 87 to 90 or 235 to 238, are difficult to understand.

Response: We are greatly thankful to this referee for pointing out this. In response to this problem, we have revised the relevant text, including the abstract and other contents in the text. Also, we have invited a native English speakers to proofread the full text and then produced the corresponding proof. For the incomprehensible sentences you mentioned, we have made careful revision to ensure that the sentences are properly corrected.

POINT 2: Authors need to reread the text to correct mistakes leading to confusion or false informations. For instance, the sentence line 103 mentions that the two modes of miRNA biological function operate in the nucleus whereas it occurs in the cytoplasma.

Response: We are grateful to the reviewer for bringing up the mistake. We are sorry for "nuclear" misused in the text. We originally wanted to express these two ways by which miRNAs function in eukaryotic cells.

POINT 3: The plan of the review needs to be rethinked in order to improve the clarity of the text. An effort of structuration should be made in order to avoid the “catalog effect” of the review.

Response: We thank you for raising this question. Now, we have adjusted the headings at all levels reasonably to make the article structure as clear and readable as possible. In fact, each of our titles is graded, but perhaps it was unified during the editing process, finally, we have renumbered the title of the paper.

POINT 4: A paragraph placing the expression of key miRNA involved in EMT process in the context of clinical assessment such as prediction of metastasis, survival etc… would be a plus.

Response: First of all, thank you very much for asking such a constructive question for us. We also agree with your view, because from a clinical point of view, the research will be more purposeful, targeted and applied. This paper describes some miRNAs with clinical experiments, including miRNA-200 family, miRNA-30 family, miRNA-34 family, miRNA-199 family, etc., all of which have been reported to be abnormally expressed in patients. As described in our perspective, the regulatory mechanism of miRNAs on EMT needs to be further combined with the clinical data.

POINT 5: The citation of previous recent studies and reviews such as Kerche et al 2022 PMID: 34981333, Garinet et al 2021 PMID: 34642464, Wei et al 2021 PMID: 33416170, Ashrafizadeh et al 2020 PMID: 32664703, would be interesting.

Response: Thank you for making this recommendation for relevant literature. Now we have revised all the relevant text by citing these publications referenced by NO. 76, 81, 119, 131 and 142.

POINT 6: The miR-199b-3p is an interesting miRNA in the field and deserve also to be cited.

Response: Thank you very much for this suggestion. The important role of miRNA-199b-3p in the process of cancer EMT has been discussed in the revised version and also cited in references 81 and 119.

Reviewer 3 Report

Feng and coworkers presnet the manuscript entitled "MicroRNA, an indispensable group of regulators in tumor Epithelial to mesenchymal transition". This a very very published topic in cancer, thus my major concern about the present manuscript is the lack of novelty.

Many reviews on the roles of microRNAs in EMT process and trasncription factors involved, have been published. Several concerns must be addressed before potential consideration of this study for revision.

2. Generation and mechanisms of canonical miRNAs. This topic have been reviewed and publsihed elsewhere, thus I found irrelevant, unless new fresh information about microRNAs biogenesis and maturation is included. Please inlcude new info in this section, different to those published in a lot or papers and reviews.

3. EMT. The same for this section.

Figure 2. This figure does not provide novel insigths. Please inlcude novel data and many more details in the proteins involved, as also this figure have been describbed in a lot of reprots.

Figure 3. Schematic diagram of EMT plasticity during metastasis of cancer cells. AThis figure is very cinfusing and incmplte. Moreover, the legend contains  many ortograpic mistakes and typos.

Finally, the description of the role of miRNA regulating EMT in the different types of  cancers, its very poor.

I recommend to focus in a specific type of cancer and deeply describe all the info.

Author Response

Thank you very much for your carefully reviewing this manuscript. All points raised by you have been addressed:

POINT 1: Generation and mechanisms of canonical miRNAs. This topic have been reviewed and published elsewhere, thus I found irrelevant, unless new fresh information about microRNAs biogenesis and maturation is included. Please include new info in this section, different to those published in a lot or papers and reviews.

Response: We are grateful for your critical comments. This mini-review only gives a clear understanding of the mode of miRNA acing on EMT in cancer, which is an interesting area.

POINT 2: EMT. The same for this section.

Response: Thank you very much for this suggestion. The basic process of EMT in development and its discovery should be introduced before we will emphasize why this process leads to tumor metastasis and invasion.

POINT 3: Figure 2. This figure does not provide novel insights. Please include novel data and many more details in the proteins involved, as also this figure have been described in a lot of reports.

Response: According to this suggestion, we have updated some new data integrated in figure 2.

Point4: Figure 3. Schematic diagram of EMT plasticity during metastasis of cancer cells. This figure is very confusing and incomplete. Moreover, the legend contains many orthographic mistakes and typos.

Response: Now, we have revised this Figure 3 as we have well done as possible. All the mistakes have been corrected in this revised version.

Point5: Finally, the description of the role of miRNA regulating EMT in the different types of cancers, its very poor.

Response: Thank you for this critical comment. This mini-review includes two main portions. In the second part, we discussed that miRNAs can exert their effects in different tumors through non-transcription factors. We would like to give a simple introduction of regulatory effects of miRNAs on EMT, but cannot introduce every tumor in depth.

Round 2

Reviewer 2 Report

The authors have adequately modified their manuscript. The text is much more structured. The review, in this present form, can be published and is useful to the field.

Note : the figure 3 is truncated on the right, do not forget to fix it.

Author Response

Thank you very much for your careful review of this manuscript. We attach great importance to your valuable suggestions on language and other content. In response to your comments and questions, we have made the following answers:

POINT 1:  the figure 3 is truncated on the right, do not forget to fix it.

Response: Thank you very much for your careful review of our resubmission and found this issue. We have repositioned the images according to the text format for easy viewing by readers.

Finally, thank you again for your valuable comments on this article.

Reviewer 3 Report

Authors have not succesfully replied all my concerns, which must be fully addressed.

POINT 1: Generation and mechanisms of canonical miRNAs. This topic have been reviewed and published elsewhere, thus I found irrelevant, unless new fresh information about microRNAs biogenesis and maturation is included. Please include new info in this section, different to those published in a lot or papers and reviews.

POINT 2: EMT. The same for this section.

Point5: Finally, the description of the role of miRNA regulating EMT in the different types of cancers, its very poor.

Authors reply is not satisfactory.

Finally, figures sizes should be adjusted to the text.

Author Response

Thank you very much for your careful review of this manuscript. We attach great importance to your valuable suggestions on figures and content. In response to your comments and questions, we have made the following answers:

POINT 1: Generation and mechanisms of canonical miRNAs. This topic have been reviewed and published elsewhere, thus I found irrelevant, unless new fresh information about microRNAs biogenesis and maturation is included. Please include new info in this section, different to those published in a lot or papers and reviews.

Response: We are very grateful for your valuable comments. We also agree with your statement, In response to your comments, We are sorry to tell you that we have not found a new generation mechanism in the existing literature. At the same time, we sincerely hope that if you have new insights, you can also provide corresponding references to enrich our manuscript. here we mainly want to introduce the classical miRNA production pathway, although it has been mentioned in other article. However, it is displayed here, mainly to give readers a basic understanding, so that readers who are interested in this direction can have a simple understanding of the mode of action of miRNA to facilitate the understanding of the following. In addition, we also noticed that there are also non-canonical pathways for the production of miRNAs, but there is no need to go into details here. Finally, we cite the original source of the pictures in the manuscript, so as to facilitate readers' better in-depth study and understanding if the reader needs.

POINT 2:EMT. The same for this section.

Response: Thank you very much for this suggestion, First of all, we admit that EMT transformation has been studied by many scholars. In view of the logical ideas of the manuscript, a brief introduction to the EMT process is required first. here, we still want to take readers to review the basic process of EMT development and its discovery process, we believe that this will be helpful to understand EMT in depth. Because in the following, we will not emphasize why the change of the marker protein will further lead to tumor metastasis and invasion. Therefore, we hope to give readers a clearer explanation in this part. Therefore, we cannot remove this part, the schematic diagram in Figure 2 will directly and clearly show such a process, which is easier for readers to understand, if the author needs in-depth understanding, we also cite the original source of the picture in the text for the convenience of readers. So we hope you can understand our intention.

Point 3: The description of the role of miRNA regulating EMT in the different types of cancers, its very poor.

Response: Thank you for this valuable suggestion, we believe that if only one tumor type is selected to describe, it will definitely lead to more in-depth research. On this point, we very much agree with your statement. But the structure of the text is mainly divided into two parts. In the first part, we try to illustrate that miRNAs can exert their effects through transcription factors. In the second part, we wanted to show that miRNAs can exert their effects in different tumors through non-transcription factors. These two parts will also have some intersection in the same type of tumor, which is unavoidable in this article. That is, miRNAs can act through transcription factors and non-transcription factors in the same tumor, so the description in the latter part is not deep enough. We want to introduce the regulatory effect of miRNA on EMT as comprehensively as possible, so we are sorry that we cannot introduce every tumor in depth.

Point 4:Finally, figures sizes should be adjusted to the text.

Response: Thank you very much for your careful review of our resubmission and found this issue. according to the text format, we have adjusted the size and position of the picture for the convenience of readers.

Finally, thank you again for your valuable comments on this article.